# Surgical Outcomes of Lensectomy-Vitrectomy with Primary Intraocular Lens Implantation in Children with Bilateral Congenital Cataracts

**DOI:** 10.3390/jpm13020189

**Published:** 2023-01-21

**Authors:** Hongzhe Li, Xiaolei Lin, Xin Liu, Xiyue Zhou, Tianke Yang, Fan Fan, Yi Luo

**Affiliations:** 1Department of Ophthalmology, Eye Hospital and School of Ophthalmology and Optometry, Wenzhou Medical University, Wenzhou 325027, China; 2Department of Ophthalmology, Eye & ENT Hospital, Shanghai Medical College, Fudan University, Shanghai 200031, China; 3Department of Ophthalmology, Shanghai Eye Disease Prevention, and Treatment Center, Shanghai Eye Hospital, Shanghai 200040, China

**Keywords:** congenital cataract, vitrectomy system, intraocular lens implantation, visual outcomes, postoperative complications, risk factors

## Abstract

In this study, we evaluated the long-term surgical outcomes of lensectomy-vitrectomy with primary intraocular lens (IOL) implantation in children with bilateral congenital cataracts (CCs) and investigated the potential risk factors for low vision. A total of 148 eyes in 74 children who underwent lensectomy-vitrectomy with primary IOL implantation were enrolled in this study. The surgery age was 44.04 ± 14.60 months, with a follow-up period of 46.66 ± 14.34 months. The final BCVA was 0.24 ± 0.32 logMAR, and low vision was found in 22 eyes (14.9%). Postoperative complications requiring additional surgeries included VAO (4 eyes, 5.4%), IOL pupillary captures (2 eyes, 2.0%), iris incarceration (1 eye, 0.7%), and glaucoma (1 eye, 0.7%). A higher incidence of VAO and larger postoperative refractive error was observed in younger children (≤2 years old) than in elder children (>2 years old) (*p* = 0.003, *p* = 0.047, respectively). Final BCVA was affected by preexisting comorbidity (*p* < 0.001), cataract density (*p* < 0.001), cataract size (*p* = 0.020), occurrence of postoperative complications (*p* = 0.011), and ASE (*p* = 0.008). Multivariate analysis showed that denser cataracts (OR = 9.303, *p* = 0.035) and preexisting comorbidity (OR = 4.712, *p* = 0.004) were the significant predictors of low vision. In conclusion, lensectomy-vitrectomy with primary IOL implantation is an effective and safe treatment for CC. The long-term visual outcome is encouraging in children with bilateral CC undergoing this procedure with a low rate of postoperative complications requiring surgeries. Moreover, eyes with denser cataracts and preexisting comorbidity may have a high risk of low vision.

## 1. Introduction

Congenital cataract (CC) is the crystalline lens opacity that presents at birth or early time in childhood [1]. It has remained the leading cause of treatable childhood low vision in the world. The global prevalence of CC is estimated as between 2.2/10,000 and 13.6/10,000 [2,3]. Macular development begins at birth and ends at 5 to 7 years old, which has been provided as the critical stage of visual development. During this period, lens opacity hinders optical transmission, causing insufficient visual stimulation, thus affecting the development of the macula and resulting in deprivation amblyopia [4,5].

With the advances in surgical techniques, surgical devices, and intraocular lenses (IOL), the success rate of CC surgery has been increasing. However, surgical management is still a challenging task because of the soft eyeball wall, elastic lens capsule, various morphologies of cataracts, and elevated inflammatory reaction due to irritation of the iris [1]. Postoperative complications are also more common in young children than adults, such as visual axial opacification (VAO), secondary glaucoma, IOL dislocation, and corectopia, which are still major obstacles to visual rehabilitation [6,7].

At present, the main treatment option for CC is phacoemulsification, posterior capsulotomy, and anterior vitrectomy with or without IOL implantation [8,9]. With the development of vitreous suction-cutting devices in recent years, soft cataracts in infants can be easily aspirated with small-gauge instruments, lensectomy by vitrectomy system has attracted the attention of surgeons because of its advantage of minimizing surgically induced trauma and inflammation thus hastening postoperative recuperation [10,11]. However, data on the result of these procedures are limited, probably as a result of the rarity of CC. Anyway, previous studies mainly focused on infants without IOL implantation and only had short-term follow-up periods [12,13,14,15]. With primary IOL implantation has become more frequent, additional reports are needed to evaluate the long-term surgical outcomes of lensectomy-vitrectomy combined with primary IOL implantation. Moreover, treatment for CC is complex and intensive, and there is great variability in outcomes following CC surgery [16,17,18]. Some achieve good postoperative visual acuity, while some still suffer from visual impairment and even blindness. Although it has been reported that numerous factors may affect postoperative visual outcomes in recent years [19,20], the crucial predictors for low vision still remain uncertain.

Therefore, the purpose of this study was to evaluate the long-term surgical outcomes of lensectomy-vitrectomy with primary IOL implantation in children with bilateral CCs and to investigate the potential risk factors for low vision.

## 2. Materials and Methods

### 2.1. Patients

We retrospectively reviewed the children with bilateral CCs who underwent lensectomy-vitrectomy with primary IOL implantation between January 2015 and December 2020 at the Eye, Ear, Nose, and Throat (EENT) Hospital of Fudan University. Children with a minimum of 1 year of follow-up after surgery were eligible for inclusion. The exclusion criteria were: (1) concurrent with other primary ocular diseases, such as aniridia, glaucoma, microcornea, microphthalmos, persistent fetal vasculature (PFV), or other fundus abnormalities; (2) history of ocular trauma, previous ocular surgery, and ocular inflammation or infection; (3) accompany with systemic diseases which could influence learning ability; (4) poor-compliance with follow-up and amblyopia therapy.

### 2.2. Surgical Technique

All surgeries were performed under general anesthesia by one experienced surgeon (Y. Luo) using the Millennium Microsurgical System (Alcon, Fort Worth, TX, USA) and the 25-gauge microincision vitrectomy system. The surgical procedures were described in our previous studies [10,11]. Briefly, a 25-gauge infusion cannula was inserted through the 4 or 8 o’clock limbal incision to maintain the anterior chamber with a balanced salt solution (BSS; Alcon). A cutting tip of the 25-gauge vitrectomy instrument was introduced through an incision at the 12 o’clock position. A central anterior capsulotomy of 5.0–5.5 mm diameter was created using the vitrector. Lens material was removed at a cutting rate of 600 cuts per minute and a maximum suction pressure of 400 mmHg (Figure 1a). A posterior capsulotomy of 4.0–4.5 mm diameter was created, followed by a limited anterior vitrectomy (Figure 1b). The microcannula at the 12 o’clock incision was then removed, and the incision was enlarged to 2.6 mm. After the ophthalmic viscosurgical device (OVD) (DisCoVisc; Alcon) was injected, a one-piece acrylic foldable IOL (AcrySof SA60AT; Alcon) was implanted into the capsular bag (Figure 1c). The limbal incision was closed with one or two 10–0 nylon sutures (Ethilon 9033; Allmedtech, Beverley Hills, CA, USA), and the corneal stroma at the limbal side port was hydrated with BSS after removal of the infusion cannula. The fellow eye was operated on a week after the first eye surgery.

The Barrett universal II formula was used to calculate the IOL power. In consideration of future axial growth, the target IOL power was chosen according to the child’s age using a scale, suggesting under-powering by +6 D at 1 year of age and decreasing 1 D per year, excepting no correction was specified for children 7 years and older (Table 1).

### 2.3. Follow-Up and Amblyopia Treatment

All children were examined postoperatively at 1 day, 1 week, and 1 month and at intervals of 3 months thereafter until the last follow-up. Each examination included a complete ophthalmological examination and monitoring of amblyopia treatment, which included refractive correction, with or without patching. The treatment period was determined to be the duration until the peak best-corrected visual acuity (BCVA) was reached.

### 2.4. Ophthalmic Examinations

Cataract morphology and eye position were examined preoperatively. Cataract density was graded and stratified into two groups according to previous studies [21,22]. Cataract free lens and cataract fully impairing the red reflex were subjectively graded as grade 0 and 10, cataracts with static morphologies were assigned fixed grades (ie, grade 1 for anterior polar and sutural cataracts, and grade 6 for posterior polar cataracts) and progressive cataracts were given grading ranges, indicating the initial and likely final morphology states [21]. Denser cataracts were defined as grade 8 or above, and less dense cataracts were defined as grade under 8 [22]. Moreover, the location and size of cataracts were also measured preoperatively based on the work of Chen et al. [23]. Briefly, cataracts not fully covering the 3 mm of the center of the lens were categorized as paracentral, while cataracts more than 3 mm from the center of the lens were categorized as central (Figure 2a). The size of the cataract was measured by ImageJ (version 1.42; Wayne Rasband, Rockville, MD, USA) on the anterior segment photograph with reference to corneal diameter (Figure 2b).

The main postoperative outcome included the final best corrected visual acuity (BCVA), refractive errors, postoperative complications, and additional surgeries. Intraocular pressure (IOP) was measured by Non-contact Tonometer (Canon). Glaucoma was defined as intraocular pressure (IOP) was 21 mmHg or higher combined with changes in the cup-to-disc ratio [24]. VAO was defined as lens material regrowth extending into the pupillary space that obscured the visual axis. Corectopia was considered when an irregular pupil was observed. Refractive error was calculated as the spherical equivalent refraction, using the algebraic power of the sphere plus half the cylindrical power. Absolute value of refractive error (ASE) was used for analysis. The BCVA was measured with a standard crowded Snellen chart and was converted to the logarithm of the minimum angle of resolution (logMAR) notation for statistical analysis. The final BCVA <20/66 Snellen (0.53 logMAR) was defined as low vision.

### 2.5. Statistical Analysis

All the statistical analyses were conducted using SPSS software (version 20.0, IBM Inc., Armonk, NY, USA). Continuous variables were presented as the mean ± standard deviation (SD). Categorical variables were presented with numbers and percentages. Because ocular characteristics and visual acuity are eye-specific, analyses were run according to the eye rather than patients. These were performed with all eyes combined by the generalized estimating equation method, which allows data from both eyes to be used while accounting for the correlation between the two eyes of a single patient. The Chi-square test and Student’s t-test were performed for comparative analyses. The relationships between the two variables were assessed with Pearson’s correlation or Spearman’s correlation. The risk factors for low vision were assessed with Multivariable logistics analysis. A *p*-value of <0.05 was considered statistically significant.

## 3. Results

A total of 148 eyes in 74 children were enrolled in this study. Among all the eyes, total cataracts accounted for the largest proportion (33.8%), followed by lamellar cataracts (25.7%), nuclear cataracts (21.6%), posterior polar cataracts (6.7%), and others. Strabismus occurred in 26 eyes (17.6%), while nystagmus occurred in 20 eyes (13.5%). The mean surgery age was 44.04 ± 14.60 months (range, 19–71 months). The mean follow-up period was 46.66 ± 14.34 months (range, 12–72 months), and the age at the last follow-up was 7.56 ± 0.75 years (range, 6.4–9.4 years) (Table 2).

Visual outcome and postoperative complications were shown and further stratified by surgery age (Table 3). The mean final BCVA was 0.24 ± 0.32 logMAR, and low vision was found in 22 eyes (14.9%). The mean ASE at the final follow-up visit was 2.07 ± 1.78 D, and it was higher in younger children (≤ 2 years old) than in elder children (>2 years old) (2.60 ± 2.06 D vs. 2.01 ± 1.74 D, *p* = 0.047). Postoperative complications requiring additional surgeries were recorded in 8 eyes (5.4%). In the early postoperative period (from one day to one month after surgery), IOL pupillary capture occurred in 2 eyes (2.0%) postoperative one week. One eye (0.7%) developed iris incarceration in the limbal incision the first day after surgery am in the late postoperative period (more than one month after surgery). VAO was observed in 4 eyes (5.4%), and the mean time of occurrence was 5.25 ± 2.63 months (range, 3–9 months) after surgery and receiving additional vitrectomy to remove it. Glaucoma was reported in 1 eye (0.7%) three months after surgery, which did not respond well to the topical antiglaucoma medication, and finally received glaucoma surgery to maintain stable IOP. The overall incidence of postoperative complications and VAO was significantly more common in younger children than in elder children (*p* = 0.029, *p* = 0.003, respectively).

In the last follow-up visit, eyes with preexisting comorbidity had worse BCVA than those without (0.50 ± 0.37 vs. 0.14 ± 0.22, *p* < 0.001), and eyes with less dense cataracts achieved better BCVA than eyes with denser cataract (0.09 ± 0.15 vs. 0.31 ± 0.34, *p* < 0.001). The final BCVA in eyes with postoperative complications was much worse than those without (0.52 ± 0.40 vs. 0.23 ± 0.30, *p* = 0.011). No significant differences were noted in the final BCVA between the different gender groups and cataract location groups (both *p* > 0.05). Both the cataract size (r = 0.191, *p* = 0.020) and ASE at the last follow-up visit (r = 0.219, *p* = 0.008) had a positive correlation with the logMAR value of the final BCVA, but not surgery age and follow-up period (both *p* > 0.05).

Compared with the non-low vision group, the low vision group had a relatively higher rate of denser cataracts (95.5% vs. 65.9%, *p* = 0.003) and preexisting comorbidity (63.6% vs. 17.5%, *p* < 0.001). It also had the greater size of cataract (15.38 ± 6.73 vs. 11.68 ± 6.08, *p* = 0.038) and larger ASE at the final follow-up visit (2.78 ± 1.63 vs. 1.94 ± 1.78, *p* = 0.035). Low vision outcome was not significantly associated with gender, cataract location, preoperative AL, Surgery age, Follow-up period, and postoperative complication (All *p* > 0.05) (Table 4). The further multivariate analysis has shown that denser cataracts (OR = 9.303, *p* = 0.035) and preexisting comorbidity (OR = 4.712, *p* = 0.004) were the significant predictors for low vision (Table 5).

## 4. Discussion

Briefly, our study demonstrated that lensectomy-vitrectomy with primary IOL implantation is an effective and safe treatment for CC. The long-term visual outcome is encouraging in children with bilateral CC undergoing this procedure with a low rate of postoperative complications requiring surgeries. Moreover, eyes with denser cataracts and preexisting comorbidity may have a high risk of low vision.

There are two approaches for performing surgery: the pars plana approach and the limbal approach. when managing infantile CC without IOL implantation, the pars plana approach provides a sufficient lensectomy and anterior vitrectomy and avoids conjunctival peritomy and sutures [12,15]. In our study, we used the limbal approach for lensectomy and lensectomy-vitrectomy, which could provide more precise manipulations under direct vision, and IOL could be directly implanted into the capsular bag through the enlarged limbal incision [10,25]. In the last follow-up visit, only 22 eyes (14.9%) remained low vision, and over 70% of eyes achieved 20/40 or better. This proportion was better than previous reports on bilateral CCs [19,26,27]. The possible reasons are as follows: (1) The introduction of microincision vitrectomy instruments minimized surgically induced trauma and inflammation and therefore hastened postoperative recuperation and enabled immediate optical correction and amblyopic treatment; (2) Longer follow-up period and older age at the last visit in our study, as visual outcomes after CC surgery in children tend to improve with time; (3) children with poor compliance with amblyopia therapy were eliminated which could obstacle the visual rehabilitation which is vital for visual rehabilitation [28].

Postoperative complications requiring additional surgeries were recorded in 8 eyes (5.4%). In the early postoperative period, two eyes (1.3%) developed IOL pupillary capture, and one eye (0.7%) developed iris incarceration in the limbal incision. In the late postoperative period, the most common complication, VAO, developed in 4 eyes (2.7%), and glaucoma was reported in 1 eye (0.7%). All of them were well-controlled after immediate surgical intervention without any recurrence. No cases of serious postoperative complications, such as retinal detachment or endophthalmitis, were observed during the follow-up period. The occurrence of postoperative complications requiring surgical intervention was lower compared with the previous studies [29,30,31]. This may be attributed to the fact that children with aniridia, congenital glaucoma, microcornea, microphthalmos, persistent primary vitreous, Marfan syndrome, history of ocular trauma, and PFV were excluded in our study, whereas these patients are always associated with a high incidence of postoperative complications. Anyway, microincision vitrectomy instruments can minimize disturbance and irritation to the iris, decreasing the intraoperative and postoperative inflammation which contributes to the formation of VAO [7], and the limbal incision could avoid trabecular meshwork injury may also contribute to a lower incidence of postoperative glaucoma. A higher incidence of VAO was observed in younger children than in elder children. This finding was similar to the results of previous studies. The IoLunder2 cohort study pointed out that increasing age at surgery was independently protective against the development of VAO [7,29]. The Infant Aphakia Treatment Study Group (IATS) also indicated that IOL implantation in children younger than 2 years of age carries a greatly increased risk of requiring secondary procedures for VAO [8,31]. Therefore, close monitoring of complications is recommended, especially for children who received surgery under 2 years of age. The mean follow-up period was 46.66 ± 14.34 months, and the age at the last follow-up was 7.56 ± 0.75 years. It may not be long enough to detect the onset of glaucoma. As children age, a higher incidence of postoperative glaucoma may be found with the extension of follow-up [32].

Our results showed that preexisting comorbidity was among the crucial predictors of low vision. Strabismus and nystagmus are well-known consequences in CC children, which result from visual deprivation during the critical period of visual development, and their presence of them has been shown to be an obstacle to visual rehabilitation [33,34]. Therefore, CC surgery was recommended before the occurrence of irreversible impairment in visual development as evidenced by strabismus and nystagmus formation, which requires early disease detection to ensure the selection of surgical timing. We also found that denser cataract was also a risk factor for low vision. Similarly, Rong indicated that worse outcomes after surgery occurred in children with total opacity cataracts [35], and Parks found that visual outcomes differed significantly by cataract type, visual acuity in the lamellar and posterior lentiglobus groups was better than in the nuclear group [36]. All these results emphasized the significant influence of cataract density on visual deprivation and amblyopia.

In addition, we found cataract size had a positive correlation with the logMAR value of the final BCVA, which means a better visual outcome was achieved in eyes with a smaller size of cataract, which was also consistent with previous reports [23,37]. Although it was not significantly associated with low vision, this parameter could be taken into consideration when making a surgical decision. Anyway, although prompt refractive correction was performed soon after surgery, the variable refractive error was still significantly related to the visual outcome in our study. This may be due to the children’s poor compliance with wearing frame glasses in their early life. Previous studies demonstrated that myopic shift is likely to occur in children who were younger than 2 years of age when they underwent IOL implantation, and earlier IOL implantation surgery meant a larger myopia drift in the future [38,39,40]. The children younger than 2 years of age in our study also had a larger ASE than those elder children (2.60 ± 2.06 D vs. 2.01 ± 1.74 D, *p* = 0.047). Therefore, IOL implantation in young children should be cautious, and future research on the myopic shift and IOL calculation formula are necessary.

Lots of previous studies emphasized the importance of early intervention [41,42]. However, surgery age did not significantly affect the final BCVA in our series. This is surprising because children with CC who are visually deprived for longer periods of time are thought to be more amblyopic. However, our study showed that denser cataracts were likely to present much earlier than less dense cataracts, perhaps negating the effect of lag time on vision. On multivariate analysis, cataract density was found to be a significant predictor of low vision, suggesting that it may be a more important determinant of deprivation amblyopia than delay in presentation.

There are some limitations in the current study. First, the retrospective design could have led to the heterogeneity of the study population. Some children without significant visual improvement may have lost follow-up due to poor compliance. Second, the small numbers of the study population, especially the younger children (≤2 years old), may have diminished the power of the current study. Additionally, as children age, ocular outcomes, especially the incidence of postoperative complications, may change. Further investigation might be necessary to follow the long-term results.

## 5. Conclusions

Lensectomy-vitrectomy with primary IOL implantation is an effective and safe treatment for CC. The long-term visual outcome is encouraging in children with bilateral CC undergoing this procedure with a low rate of postoperative complications requiring surgeries. The monitoring of postoperative complications and refractive error is warranted, especially for children under 2 years of age. Numerous factors may determine the final visual acuity, among which denser cataracts and preexisting comorbidity were the significant predictors of low vision.

## Figures and Tables

**Figure 1 jpm-13-00189-f001:**
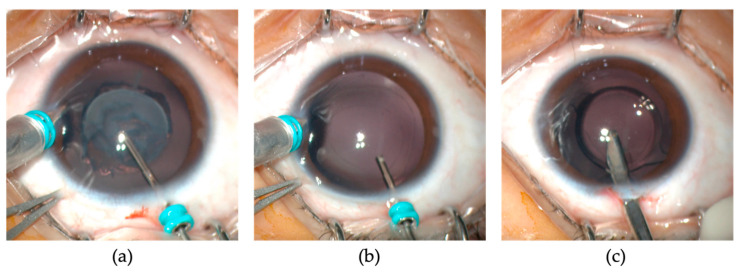
(**a**) Lensectomy was performed after a central anterior capsulotomy of 5.5 mm diameter using a 25-gauge vitrectomy cutter via a limbal incision. A limbal port incision was made for infusion to maintain the anterior chamber. The eye was positioned using a pair of forceps. (**b**) Anterior vitrectomy after a posterior capsulotomy of 4.5 mm diameter. (**c**) The 12 o’clock limbal incision was enlarged, and a one-piece acrylic foldable IOL was implanted into the capsule bag.

**Figure 2 jpm-13-00189-f002:**
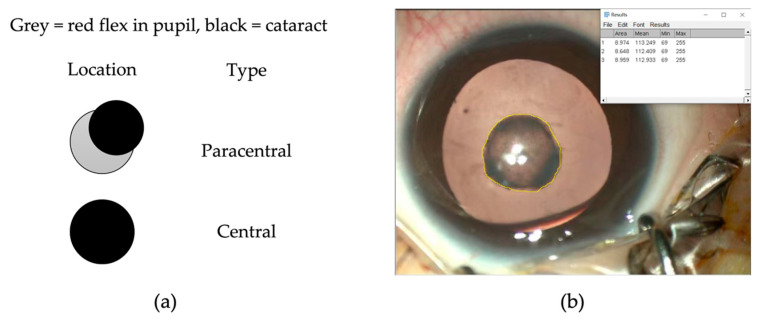
(**a**) Cataract-impairment of red reflex. The location of the cataracts was categorized as central or paracentral. Cataracts not fully covering 3 mm of the center of the lens were categorized as paracentral, while cataracts more than 3 mm from the center of the lens were categorized as central. (**b**) The size of the cataract was measured by ImageJ on the anterior segment photograph with reference to corneal diameter.

**Table 1 jpm-13-00189-t001:** Target hyperopia diopter of target IOL power chosen according to the child’s age.

Age at IOL Implantation (Years)	Target Hyperopia Diopter (D)
1 ~ 2	+5 ~ +6
2 ~ 3	+4 ~ +5
3 ~ 4	+3 ~ +4
4 ~ 5	+2 ~ +3
5 ~ 6	+1 ~ +2

IOL: intraocular lens, D: diopter.

**Table 2 jpm-13-00189-t002:** Demographic of the study population.

Characteristics	Mean ± SD or N (%)
Patients/eyes	74/148
Male/female	44/30
Cataract morphology	
Total	50 (33.8)
Nuclear	32 (21.6)
Lamellar	38 (25.7)
Posterior polar	10 (6.7)
Other types	18 (12.2)
Preexisting comorbidity	
strabismus	26 (17.6)
nystagmus	20 (13.5)
Preoperative AL (mm)	21.55 ± 1.45 (19.14–26.70)
Surgery age (months)	44.04 ± 14.60 (19–71)
Follow-up (months)	46.66 ± 14.34 (12–72)
Age at last follow-up (years)	7.56 ± 0.75 (6.4–9.4)

SD: standard deviation, N: number, AL: axial length.

**Table 3 jpm-13-00189-t003:** Visual outcome and postoperative complications.

Characteristics	All (n = 148)	Younger Children (≤2 Years Old) (n = 14)	Elder Group (>2 Years Old) (n = 124)	*p* Value
BCVA (logMAR)	0.24 ± 0.32	0.26 ± 0.30	0.24 ± 0.32	0.834
Low vision, n (%)	22 (14.9)	3 (27.2)	19 (15.8)	0.439
Refractive error at last follow-up (ASE, D)	2.07 ± 1.78	2.60 ± 2.06	2.01 ± 1.74	0.047 *
Postoperative complications requiring surgeries, *n* (%)	8 (5.4)	3 (21.4)	5 (4.0)	0.029 *
Early postoperative complications				
IOL pupillary capture	2 (1.3)	0 (0.0)	2 (1.4)	0.819
Iris incarceration in incision	1 (0.7)	0 (0.0)	1 (0.8)	0.905
Late postoperative complications				
VAO	4 (2.7)	3 (21.4)	1 (0.8)	0.003 *
Glaucoma	1 (0.7)	0 (0.0)	1 (0.8)	0.905

BCVA: best corrected visual acuity, ASE: absolute of spherical equivalent, D: diopter, VAO: visual axial opacification, IOL: intraocular lens. * denotes statistically significant.

**Table 4 jpm-13-00189-t004:** Differences in clinical features between the low vision group and non-low vision group.

Characteristics	Low Vision Group (n = 22)	Non-Low Vision Group (2 = 126)	*p* Value
Gender (male: female)	12:10	76:50	0.389
Cataract density (grade ≤7: grade >7)	1:21	43:83	0.003 *
Cataract location (central: paracentral)	7:15	53:73	0.254
cataract size (mm^2^)	15.38 ± 6.73	11.68 ± 6.08	0.038 *
Preoperative comorbidity (yes: no)	14:8	22:104	<0.001 *
Preoperative AL (mm)	22.00 ± 1.53	21.47 ± 1.42	0.114
Surgery age (months)	45.55 ± 14.17	43.78 ± 14.72	0.241
Follow-up period (months)	45.16 ± 12.37	46.92 ± 14.68	0.136
Refractive error (ASE, D)	2.78 ± 1.63	1.94 ± 1.78	0.035 *
postoperative complications (yes: no)	3:19	5:121	0.097

ASE: absolute of spherical equivalent; D: diopter. * denotes statistically significant.

**Table 5 jpm-13-00189-t005:** Risk Factors for low vision: Multivariate Analysis According to Final Model.

Characteristics	OR	95% CI	*p* Value
Cataract density			
grade ≤ 7	-	-	-
grade > 7	9.303	1.171–73.903	0.035 *
cataract size (mm2)	1.033	0.978–1.092	0.247
Preexisting comorbidity			
no	-	-	-
yes	4.712	1.656–13.410	0.004 *
Refractive error (ASE, D)	1.154	0.891–1.494	0.277

ASE: absolute of spherical equivalent; D: diopter. * denotes statistically significant.

## Data Availability

Not applicable.

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
