# Peer review of "Surgical Outcomes of Lensectomy-Vitrectomy with Primary Intraocular Lens Implantation in Children with Bilateral Congenital Cataracts"

_jpm, 2023, doi:10.3390/jpm13020189_

Round 1

Reviewer 1 Report

Line 68 – missing word

Line 152 – error with IOLunder2

Line 248 – it is difficult to understand why the authors would have lower complication rates than other reports, since their technique is not different from established surgical techniques that have been reported elsewhere (PMID 35307324).

Line 285 – it is also difficult to understand why this result is different from prior literature. It does not make much sense why the low vision outcome was not associated with surgery age.

Authors need to acknowledge that their follow up period may not be long enough to detect the onset of glaucoma.

Reviewer 2 Report

In this study the authors present the long-term surgical outcomes of lensectomy-vitrectomy

with primary IOL implantation in children with bilateral congenital cataracts. 

The paper is well written, but some corrections must be done.

 ·      Line 48: “IOL dislocation and et al”: is something missing?

·      Line 62:  Some achieve

·      Line 93. “a foldable IOL”. Please describe the IOL (or IOLs) used.

·      Lines 98 -99 “SRK-T, Holladay 1, Hoffer Q, Haigis and Barrett Universal II formulae were used to 98

·      calculate the IOL power”:  The authors used 5 different formulas to calculate IOL in the study. How do they choose the most correct formula for each patient? It must be clarified.

·      Lines 143-144: “VAO … was treated with an yttrium-aluminum-garnet (YAG) laser or vitrectomy”. How did the authors choose the surgical approach? How many YAG laser vs vitrectomy capsulotomy”?

·      Lines 159-160 “The final BCVA < 20/66 Snellen (0.53 logMAR) was defined as the low vision” must be placed in the “Ophthalmic evaluations” paragraph.

·      The authors divided the patients into two groups “elder children” and “younger children”, but the number of patients is very different between them (124 vs 14). Isn't the difference too big to make a statistical comparison? Moreover, the author should provide mean age±DS of each group.

·      The reference “Singh R, Barker L, Chen SI, Shah A, Long V, Dahlmann-Noor A. Surgical interventions for bilateral congenital cataract in children aged two years and under. Cochrane Database Syst Rev. 2022 Sep 15;9(9):CD003171.” must be added in the bibliography and commented in the text.

Round 2

Reviewer 1 Report

The authors have responded appropriately to reviewer concerns.

Reviewer 2 Report

The changes were performed by the authors as asked